# Spatiotemporal Dynamics of NDVI, Soil Moisture and ENSO in Tropical South America

**Diana M. Álvarez \*** and **Germán Poveda** 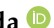

Department of Geosciences and Environment, Universidad Nacional de Colombia, Medellín 050034, Colombia; gpoveda@unal.edu.co
\* Correspondence: dialvarezm@unal.edu.co

**Abstract:** We evaluated the coupled dynamics of vegetation dynamics (NDVI) and soil moisture (SMOS) at monthly resolution over different regions of tropical South America and the effects of the Eastern Pacific (EP) and the Central Pacific (CP) El Niño–Southern Oscillation (ENSO) events. We used linear Pearson cross-correlation, wavelet and cross wavelet analysis (CWA) and three nonlinear causality methods: ParrCorr, GPDC and PCMCIplus. Results showed that NDVI peaks when SMOS is transitioning from maximum to minimum monthly values, which confirms the role of SMOS in the hydrological dynamics of the Amazonian greening up during the dry season. Linear correlations showed significant positive values when SMOS leads NDVI by 1–3 months. Wavelet analysis evidenced strong 12- and 64-month frequency bands throughout the entire record length, in particular for SMOS, whereas the CWA analyses indicated that both variables exhibit a strong coherency at a wide range of frequency bands from 2 to 32 months. Linear and nonlinear causality measures also showed that ENSO effects are greater on SMOS. Lagged cross-correlations displayed that western (eastern) regions are more associated with the CP (EP), and that the effects of ENSO manifest as a travelling wave over time, from northwest (earlier) to southeast (later) over tropical South America and the Amazon River basin. The ParrCorr and PCMCIplus methods produced the most coherent results, and allowed us to conclude that: (1) the nonlinear temporal persistence (memory) of soil moisture is stronger than that of NDVI; (2) the existence of two-way nonlinear causalities between NDVI and SMOS; (3) diverse causal links between both variables and the ENSO indices: CP (7/12 with ParrCorr; 6/12 with PCMCIplus), and less with EP (5/12 with ParrCorr; 3/12 with PCMCIplus).

**Keywords:** NDVI; soil moisture; ENSO; South America; linear correlations; wavelet analysis; nonlinear causality

## 1. Introduction

It is difficult to overstate the role of land-vegetation-atmosphere interactions on the spatiotemporal dynamics of water, energy and carbon balances in tropical South America and the Amazon River basin [1–7], more so at interannual timescales which are mainly controlled by the occurrence of El Niño–Southern Oscillation (ENSO) in the tropical Pacific [8–19]. Soil moisture plays an important role in land surface-vegetation-atmosphere interactions over the region [20–23]. Particularly, the study of Bruno et al. [24] showed that the relation between soil moisture and vegetation activity (evapotranspiration) exhibit clear-cut diurnal and seasonal cycles, and that the upper 2 m of soil supplied 56% (28%) of the water used for evapotranspiration during the wet (dry) season and that the zone of active water withdrawal reached to at least 10 m. On the other hand, the work by Zanin and Satyamurty [25] found that the amount of water precipitated over southeastern Amazonia during the autumn season influences the amount of precipitation during the winter season over the central-western region of the La Plata River basin, via a suite of soil moisture-vegetation-atmosphere feedbacks. Furthermore, the study of Negrón-Juárez et al. [26] used

detailed soil moisture data in the Amazon and showed its strong variability (with depth and time), evidencing the importance of soil texture, root uptake depths and precipitation rates, from specific events to seasonal timescales. Additionally, the work by Llopart et al. [27] showed strong differences between two different land surface parameterizations of the RegCM4 model on the simulated climate and its variability over South America (SA), in terms of the coupling between precipitation, soil moisture, evapotranspiration and sensible heat flux; and Broedel et al. [28] found that more than 40% of the total demand for transpiration is supplied by the first top meter of the soil, and that during the extreme Amazon drought of 2005 deep root uptake occurred at greater depths as a mechanism to cope with prolonged droughts.

In turn, the study by Cho et al. [29] examined the impact of deforestation on soil moisture in Rondônia, southwest (SW) Amazonia, and found a significant increase in soil moisture dryness in the deforested Amazon, in association with changes in precipitation and vegetation conditions. Additionally, using eddy covariance flux measurements across the Amazon River basin, Hasler and Avissar [30] studied the spatiotemporal variability and annual cycles of evapotranspiration (ET) in the region, and showed a strong seasonality in ET for sites near the equator (2°–3°S), with ET increasing during the dry season (June–September) and decreasing during the wet season (December–March), in phase with the annual cycle of net radiation. For stations farther south (9°−11°S) no clear seasonality was identified in either net radiation or ET [30]. Additionally, the study by Maeda et al. [31] found that both annual mean and seasonality in ET are driven by a combination of energy and water availability, and in southern Amazonian basins, despite seasonal rainfall deficits, deep root water uptake allowed increasing rates of ET during the dry season, and also that two different models could provide a consistent representation of ET seasonal patterns across the Amazon.

On the other hand, modelling studies by Mu et al. [32] showed that during the 2005 and 2010 Amazon droughts, moisture supply decreased from oceans and non-forested areas, but they were compensated by a stable supply from forests, and a relative insensitivity of forest ET to droughts, highlighting the role of forests to mitigate droughts, and the impacts of deforestation. In this regard, different studies have evaluated the seasonal and interannual variability of carbon and water fluxes in the Amazon: Manoli et al. [33] found that the decrease in Leaf Area Index (LAI) at the end of the wet season reduces ET, thus saving soil water for the upcoming dry months, and little evidence of soil moisture stress in most of the locations with ET fluxes supported by deep root water uptake. Additionally, Huete et al. [34] found a basin-wide enhanced rain forest activity (greening up) in the sunnier dry season, suggesting that sunlight may exert more influence than rainfall on rain forest phenology and productivity.

Additionally, Harper et al. [35] investigated how the avoidance of dry season water stress can increase moisture recycling and mitigate drought intensity, and Restrepo-Coupé et al. [36] investigated the seasonal patterns of Amazonian forest photosynthetic activity, and the effects of climate variability and land use, by integrating data from a network of ground-based eddy flux towers in Brazil, and found that high or increasing levels of photosynthetic activity were observed in these equatorial forests during the dry season, together with no water limitation observed in the seasonality of the ratio of sensible to latent heat flux (the Bowen ratio). These results provided strong evidence against the common idea that water limitation constrains photosynthesis in these forests, and that equatorial forests can grow leaves in the dry season, when surface solar radiation peaks. However, Samanta et al. [37] investigated the relation between the severity of drought and the spatial extent or magnitude of greening during the severe 2005 drought and found no co-variability between the severity of drought and the spatial extent and magnitude of greenness changes of Amazon forests in 2005.

The present study aims to investigate the relations between monthly series of soil moisture and vegetation dynamics (NDVI) over different regions of tropical South America and the Amazon River basin and their variability at interannual timescales, mainly con-

trolled by the El Niño–Southern Oscillation (ENSO) and its different types: Central Pacific (CP) and Eastern Pacific (EP) events [38]. The study is distributed as follows: Section 2 presents materials and methods including study area, data sets and ENSO indexes and the methodologies employed. Section 3 presents the results obtained using both raw and standardized data sets, whereas Section 4 discusses the main results, and Section 5 present the main conclusions of the study.

## 2. Materials and Methods

### 2.1. Study Area: Tropical South America

Six hydroclimatologically singular and distinctive neighboring regions within a continental-scale area in tropical South America were selected, including the Amazon River basin owing to its fundamental role on the hydroclimatic dynamics at continental and global scales [39–43]. Within the study area, the following six regions of tropical South America were defined: Colombia, Venezuela, and Western, Central, Southern and Eastern Amazonia. See Table 1 and Figure 1 for details.

**Table 1.** Location of the regions of tropical South America included in the present study.

| Region | Code | Longitude | Latitude |
|---|---|---|---|
| Colombia | CO | 77°W–70°W | 0–11°N |
| Venezuela | VE | 70°W–60°W | 0–11°N |
| Western Amazonia | WA | 76°W–67°W | 7°S–0 |
| Central Amazonia | CA | 67°W–57°W | 7°S–0 |
| Eastern Amazonia | EA | 57°W–48°W | 10°S–2°N |
| Southern Amazonia | SA | 67°W–57°W | 15°S–7°S |

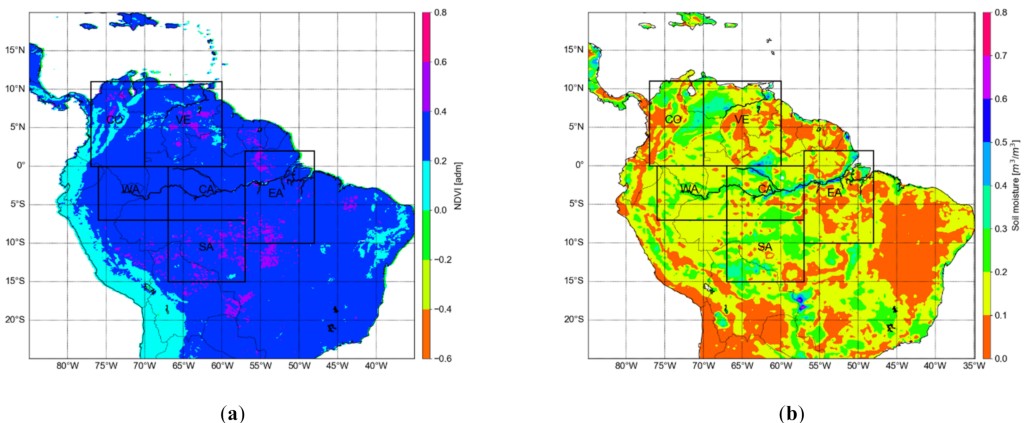

**Figure 1.** Long-term mean maps of: (**a**) normalized difference vegetation index (NDVI); (**b**) soil moisture (SMOS).

### 2.2. Data Sets

#### 2.2.1. Soil Moisture (SMOS)

Soil moisture monthly data were obtained from the Soil Moisture and Ocean Salinity (SMOS), satellite Level 3 research product, available at the Barcelona Expert Center (http://bec.icm.csic.es/land-datasets/, accessed on 25 August 2019), between 2010 and 2018. SMOS acquires temperature brightness data at 1.4 GHz which is a function of the emissivity, and hence soil moisture near the surface (approximately 5 cm) is obtained [44,45].

#### 2.2.2. Normalized Difference Vegetation Index (NDVI)

NDVI measures vegetation activity and it is estimated as the relation between (NIR-Red) and (NIR + Red) where "NIR" is the radiation reflected in the near infrared and "Red" is the radiation reflected in the red band [46,47], derived from the Advanced High-

Resolution Radiometer (AVHRR). Daily NDVI data are obtained from the Climate Data Record (CDR) at 0.05° × 0.05° spatial resolution from 1981 to 2018.

The NDVI data set compared very well with already validated and well-established MODIS products. In fact, the study of Franch et al. [48] found a good performance and consistency between the AVHRR and MODIS products, in particular, with respect to atmospheric and water vapor correction methods, but also with respect to error residuals of the Bidirectional Reflectance Distribution Function, and also to systematic calibration biases during the year.

Figure 1 shows the long-term mean maps of NDVI and SMOS over northern South America and the location of the study regions, and Table 2 summarizes the data used in this study, including the temporal and spatial resolutions. To carry out the proposed analyses, daily data are used to estimate monthly mean values within the six study areas, as shown on Table 1; such monthly temporal resolution of data allows to quantify the conjoint dynamics of NDVI and SMOS at annual and interannual (ENSO) timescales.

**Table 2.** Description of the data sets used in this study.

| Database | Variable | Resolution | | Record Period | Analysis Period | Temporal Resolution |
|---|---|---|---|---|---|---|
| | | Spatial | Temporal | | | |
| SMOS-BEC | Soil moisture | 25 km | Daily | Since 1 June 2010 | 1 January 2010 to | Monthly |
| NOAA | NDVI | 0.05° | Daily | Since 24 June 1981 | 31 December 2018 | Monthly |

### 2.2.3. El Niño–Southern Oscillation (ENSO)

El Niño–Southern Oscillation (ENSO) is the main modulator of natural climate variability at interannual timescales worldwide. It results from nonlinear interactions between the ocean and the atmosphere over the Equatorial Pacific and involves a warm phase (El Niño) and a cold phase (La Niña) in terms of sea surface temperature anomalies (SST) [49]. The Niño 3 (5°N–5°S, 150°W–90°W) and Niño 4 (5°N–5°S, 160°E–150°W) regions are shown in Figure S1, and their respective indices refer to monthly normalized SSTs.

Recent observations show that the major SST anomalies during ENSO events are frequently confined to the central Pacific (CP), distinct from the eastern Pacific (EP) warming pattern during El Niño canonical events [38,50]. The EP events show a significant periodicity between three and five years, whereas the CP events show a shorter periodicity between two and four years [50]. Diverse studies have shown that the CP event has a significant impact on global climate [38], being particularly strong in South America [51].

Due to these distinctive differences, several indices have been proposed to distinguish the two different types of El Niño; according to Sullivan et al. (2016) [50], indices for both types of El Niño are estimated as,

$$EP = Niño3\_normalised - 0.5 \times Niño4\_normalised, \qquad (1)$$

$$CP = Niño4\_normalised - 0.5 \times Niño3\_normalised, \qquad (2)$$

### 2.3. Methodology

The methodologies employed were applied to both raw and standardized times series of SMOS and NDVI over the studied regions, the latter used to study the effects of interannual variability associated with ENSO, as discussed next (see Figure S2):

- Raw time series correspond to the observed SMOS and NDVI monthly series without any manipulation, as mentioned above.
- Standardized time series of the regionally averaged values are estimated such that the anomalies, z(t), are computed by subtracting the sample monthly mean of the raw data, x(t), and dividing by the corresponding sample monthly standard deviation [52].

2.3.1. Exploratory Analysis and Linear Correlations

An exploratory analysis of the data was initially performed to quantify the behavior of both variables in terms of the mean and median, and other parameters such as dispersion, location and symmetry.

For both raw and standardized times series of the regionally averaged values of NDVI and SMOS, as well as for time series of both variables on a pixel-by-pixel scale, linear Pearson's lagged cross-correlation analyses were estimated. This method is often used as an abbreviated, single-valued measure of linear association between two variables. The following are the temporal/spatial analyses performed according to the variables correlated:

- *NDVI and SMOS.* Temporal and spatial linear Pearson's lagged cross-correlations are estimated between raw and standardized time series of NDVI and SMOS, for the regionally averaged values and on a pixel-by-pixel scale.
- *Correlations with both ENSO types.* Temporal and spatial Pearson's lagged correlations between both variables and the EP and CP indices are estimated for both raw and standardized data sets.

2.3.2. Wavelets and Cross-Wavelets Analyses

The wavelet transform is useful to study temporal variability of spectral properties of time series and the temporal cross-coherence between the phases of stationary and non-stationary series [53,54]. For the present study, we estimated the wavelet transforms of the monthly (raw and standardized) series of NDVI and soil moisture, as well as the wavelet coherence and the cross-wavelet transform between both variables. These methods allow to identify the most significant epochs and frequencies explaining the largest portion of the variance of both variables, but also those in which both two time series are in and out of phase. For estimation purposes, we used the continuous wavelet transform (CWT) [54], the cross-wavelet analysis (CWA) [55] involving the wavelet cross-spectrum (WCS) and the wavelet coherency analysis (WCO). The WCS is defined as the expected value of the product of the single wavelet power spectra of both time series at a certain time, $t_i$, on a scale $s$, and the WCO is a normalized time and scale resolved measure for the relationship between two time series, defined as the amplitude of the WCS (wavelet cross-spectrum) normalized to the two single WPS (wavelet power spectrum) [55].

We used the Morlet complex wavelet that consists of a plane wave modulated by a Gaussian, which exhibits diverse advantages with respect to other wavelets [56]. For instance, it provides a fine balance between time and frequency localization, and its Fourier period is almost equal to the wavelet scale. As such, it has been long used in the study of climatic time series [56]. Its implementation was made using *waipy*, a python package developed by Mabel C. Costa, available at https://github.com/mabelcalim/waipy, accessed on 20 January 2020.

2.3.3. Nonlinear Causality Analyses

We used time-lagged causal inference methods to quantify diverse nonlinear causality measures between monthly series of NDVI and SMOS. To that end, we used the PCMCI method that allows to identify common causal drivers and links among high dimensional simultaneous and time-lagged variables, by combining a PC Markovian condition-selection step, named after their creators [57,58], and the Momentary Conditional Independence (MCI) test. PCMCI has been applied to infer nonlinear causalities between diverse bio-geophysical phenomena [59–63] using diverse tests including linear partial correlations (ParCorr) and three types of nonlinear independence tests: GPDC, CMI and PCMCIplus: (1) GPDC is based on a Gaussian process regression and a distance correlation test on the residuals, which is adequate for a large class of nonlinear dependencies with additive noise; (2) CMI is a nonparametric test based on a *k*-nearest neighbor estimator of conditional mutual information that allows capturing almost any type of dependency [61–63]; (3) PCMCIplus can identify the full, lagged and contemporaneous causal graphs (up to the

Markov equivalence class for contemporaneous causality) under the standard assumptions of Causal Sufficiency, Faithfulness and the Markov condition. For implementation purposes, we used the *Tigramite 4.2* python package, which allows to reconstruct graphical models (conditional independence graphs) from discrete or continuously valued time series based on the PCMCI method [61]. The package has been developed by Jakob Runge and is available at https://github.com/jakobrunge/tigramite/, accessed on 20 March 2020.

These methods have been shown to be a superior tool than the traditional Fourier analysis to investigate the dynamics of climatic and biophysical data, given that the time resolution is intrinsically and optimally adjusted to the different time scales involved. This is more relevant regarding the multivariate analysis or the coupling among two variables at different frequencies throughout time, and therefore the need for a bivariate extension of the wavelet analysis [54].

## 3. Results

### 3.1. Raw Data

#### 3.1.1. Exploratory Analysis

Table 3 presents results of diverse statistical parameters of location (mean and median), dispersion (standard deviation) and symmetry (IQR, skewness, and YK) for NDVI and SMOS in the six study regions. For both variables, location indices are similar; therefore, it can be said that both series lack extreme values. According to the Inter-Quartile Range (IQR) of NDVI, most of the data are between 0.04 and 0.11 for vegetation activity, and between 0.04 and 0.07 $m^3/m^3$ for soil moisture.

**Table 3.** Analysis of location (mean and median), dispersion (standard deviation) and symmetry indices for NDVI and SMOS in the six studied regions.

| Index | NDVI | | | | | | SMOS | | | | | |
|---|---|---|---|---|---|---|---|---|---|---|---|---|
| | CO | WA | CA | SA | EA | VE | CO | WA | CA | SA | EA | VE |
| Mean | 0.25 | 0.27 | 0.30 | 0.33 | 0.32 | 0.28 | 0.16 | 0.17 | 0.19 | 0.18 | 0.13 | 0.16 |
| Median | 0.25 | 0.27 | 0.29 | 0.29 | 0.30 | 0.28 | 0.18 | 0.17 | 0.19 | 0.18 | 0.13 | 0.16 |
| Std. deviation | 0.03 | 0.04 | 0.05 | 0.09 | 0.07 | 0.04 | 0.04 | 0.03 | 0.05 | 0.04 | 0.03 | 0.04 |
| IQR | 0.04 | 0.05 | 0.09 | 0.11 | 0.11 | 0.05 | 0.05 | 0.04 | 0.07 | 0.08 | 0.05 | 0.07 |
| Skewness | 0.22 | 0.43 | 0.56 | 1.09 | 0.66 | −0.13 | −0.54 | −0.15 | 0.30 | 0.31 | 0.39 | 0.12 |
| YK | −0.07 | 0.17 | 0.32 | 0.52 | 0.42 | 0.03 | −0.47 | 0.01 | −0.32 | 0.13 | 0.11 | −0.07 |

As for symmetry, the skewness coefficient and the Yule-Kendall (YK) index for both variables indicate that the data have positive symmetry (right tail).

#### 3.1.2. Long-Term Mean Annual Cycles and Time Series

Figure 2 presents the long-term mean annual cycles of NDVI and soil moisture in the study regions. It is worth noticing that the maximum and minimum values of SMOS tend to lead those of NDVI by 2–4 months, depending on each region. The annual cycles of both variables exhibit different phases (months associated with peak values), and in some regions tend to be in phase, whereas in other regions tend to be out of phase.

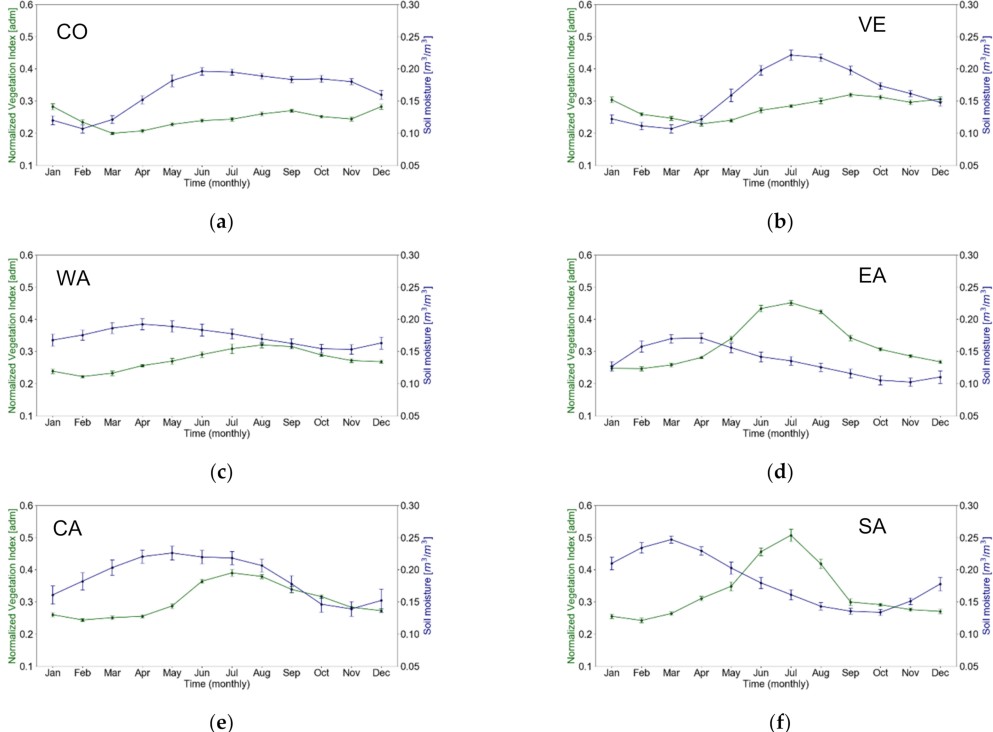

**Figure 2.** Long-term mean (2010–2018) annual cycles of NDVI and soil moisture in the six studied regions. Error bars denote the standard error of the mean. (**a**) CO: Colombia, (**b**) VE: Venezuela, (**c**) WA: West Amazonia, (**d**) EA: East Amazonia, (**e**) CA: Center Amazonia, and (**f**) SA: South Amazonia.

The simultaneous evolution of the monthly values of both variables over the studied regions are shown in Figure S3 of the Supplementary Materials. Those plots also confirm that the maximum and minimum monthly values of NDVI lag those of SMOS, with the temporal lag varying among regions, pointing out that the association (causality) SMOS to NDVI is much more significant than vice versa, as shown in the following sections.

Results also showed that NDVI peaks when soil moisture is transitioning from maximum to minimum monthly values, which points out the role of soil moisture on the hydrological dynamics of the identified Amazonian greening up Amazonia during the dry season, which deserves further investigation.

### 3.1.3. Correlation Analysis

Lagged cross-correlations between time series. Results of lagged cross-correlations between time series of regionally averaged values of both variables are shown in Figure 3, with the values of the highest correlations and the corresponding time lags summarized in Table 4. These results confirm that maximum and minimum values of SMOS lead those of NDVI by 2 months in CO and VE, by 3 months in WA, CA and EA, and by 4 months in SA. On the other hand, NDVI tend to lead SMOS at much larger time scales, as follows: by 8 months in WA and CA, by 9 months in EA and SA, and by 10 months in CO and VE.

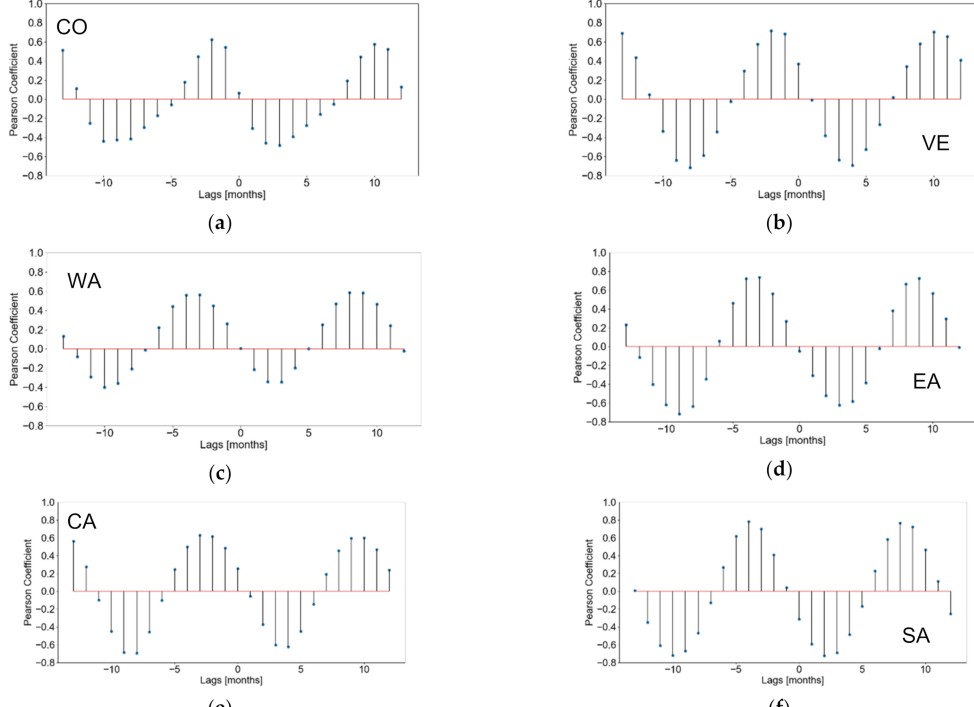

**Figure 3.** Lagged cross-correlations between monthly series of regionally averaged values of NDVI and SMOS over the studied regions. Negative lags denote SMOS leading NDVI, and positive lags denote NDVI leading SMOS. (**a**) CO: Colombia, (**b**) VE: Venezuela, (**c**) WA: West Amazonia, (**d**) EA: East Amazonia, (**e**) CA: Central Amazonia, and (**f**) SA: South Amazonia.

**Table 4.** Maximum correlation coefficients between the monthly series of regionally averaged values of NDVI and SMOS, and corresponding time lags (months), for the study regions.

| Region | SMOS Leads NDVI | | NDVI Leads SMOS | |
|---|---|---|---|---|
| | Max. Correlation | Time-Lag (Months) | Max. Correlation | Time-Lag (Months) |
| CO | 0.62 [a] | −2 | 0.57 [a] | 10 |
| VE | 0.72 [a] | −2 | 0.70 [a] | 10 |
| WA | 0.56 [a] | −3 | 0.59 [a] | 8 |
| CA | 0.75 [a] | −3 | −0.78 [a] | 8 |
| SA | 0.78 [a] | −4 | 0.76 [a] | 9 |
| EA | 0.74 [a] | −3 | 0.72 [a] | 9 |

Significance levels: [a] 99%.

- Spatial lagged cross-correlations. Figure 4 shows maps of spatial (pixel-by-pixel) lagged cross-correlations between the monthly series of NDVI and SMOS. Results indicate significant positive correlations for most regions, especially when SMOS leads NDVI by 1, 2, and 3 months, in particular over the Colombian-Venezuelan Llanos of the Orinoco River basin, and smaller negative correlations when NDVI leads SMOS by 2 and 3 months.

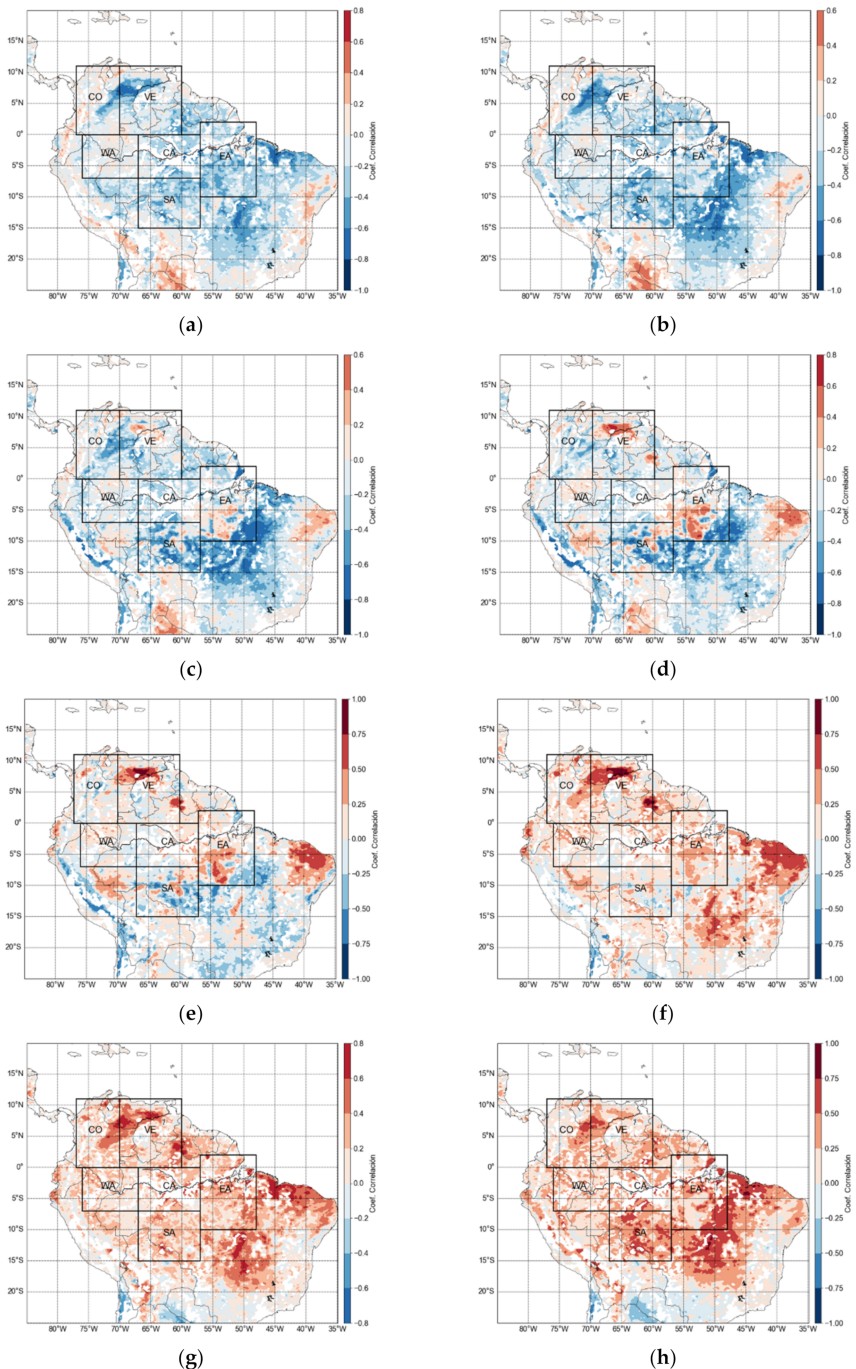

**Figure 4.** Maps of pixel-by-pixel lagged cross-correlations between NDVI and SMOS. Results show NDVI leading SMOS by: (**a**) 4 months; (**b**) 3 months; (**c**) 2 months; (**d**) 1 month; (**e**) 0 (simultaneous). Results for SMOS leading NDVI by: (**f**) 1 month; (**g**) 2 months; and (**h**) 3 months.

### 3.1.4. Wavelet Transforms, Wavelet Cross-Spectra, and Coherency Analyses

Figure 5 shows the continuous wavelet transforms of both NDVI and SMOS raw monthly series for EA and SA, respectively. There is a clear-cut common feature in the wavelet spectra of both time series, namely the significant peak at the 12-month frequency band throughout the entire record length, which becomes more evident in the Fourier and the wavelet spectra. Time series of SMOS also exhibit an important spectral peak at the 64-month frequency band, mostly associated with ENSO, albeit located inside the non-significant cone of influence. Such a 64-month spectral peak only appears in time series of NDVI for the Western Amazon (WA). The strongest intra-annual variability occurs

in Colombia, mostly due to the semi-annual cycle associated with the passage of the ITCZ, twice a year, mainly over the Andean region [64,65].

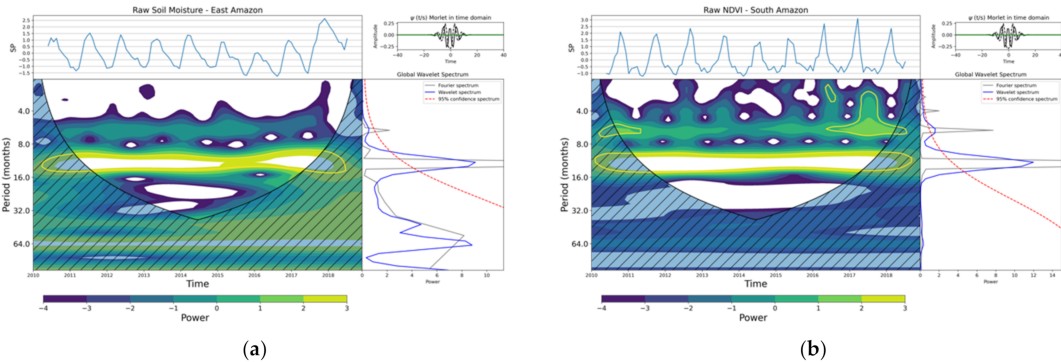

(**a**)　　　　　　　　　　　　　　　(**b**)

**Figure 5.** Wavelet analyses of the raw monthly series of NDVI and SMOS indicating that the highest portion of the variance is located in the 12-month frequency band: (**a**) SMOS—EA region; (**b**) NDVI— SA region.

From Figure 5 and Table 5 it is also worth noting that the power of the 12-month frequency band of NDVI is stronger in the EA region, followed by the CA, WA, SA, VE and CO, whereas for SMOS the 12-month frequency band explains a higher portion of the variance in the following order: SA, VE, CO, EA, CA, and WA. In turn, the 64-month frequency band (ENSO) is stronger in the following order: WA, CA, CO, EA, VE, and SA. The WA region exhibits the highest portion of the variance explained by the combination of both the 12-month and the 64-month frequency bands for NDVI and SMOS. Detailed results of the wavelet transforms are shown in Figures S4 and S5 for SMOS and NDVI, respectively.

**Table 5.** Power of the 12-month (annual cycle) and the 64-month (ENSO) frequency bands.

| Region | NDVI | | Soil Moisture | |
|---|---|---|---|---|
| | Power at 12 Months | Power at 64 Months | Power at 12 Months | Power at 64 Months |
| CO | 6 | 0 | 10 | 10 |
| VE | 10 | 0 | 12 | 7 |
| WA | 12 | 5 | 5 | 24 |
| CA | 14 | 0 | 8 | 14 |
| SA | 12 | 0 | 13 | 5 |
| EA | 15 | 0 | 9 | 9 |

Results of the wavelet cross-spectra (WCS), and the wavelet coherency analyses (WCO) are illustrated in Figure 6. The WCS quantifies the local relative phase between two series in the time-frequency space [66], as shown on the left panel. In those figures, arrows pointing to the right (3 h) indicate that both time series are in phase, whereas arrows pointing to the left (9 h) indicate that both time series are out of phase. Similar behaviors are observed in CO, VE and CA, with strong positive co-variability at frequencies associated with the annual cycle (12 months), especially from 2011 to 2017, with small time lags (1–2 months) between SMOS and NDVI, as evidenced by the arrows pointing to the 1 and 2 h. A different behavior is observed in the EA and SA regions. For the former one, there is a negative co-variability of both variables around the 12-month frequency band with a small lag (1–2 months), and a short period of time between 2015 and 2017 exhibiting a positive co-variability of both signals in the frequency bands associated with 9–14 months, albeit not completely in phase given that the arrow points to the 12 h, equivalent to a 3-month time lag between SMOS and NDVI. The behavior in SA is completely different compared with the other regions as per the strong co-variability (in phase) of both signals at intra-annual

timescales, whereas a negative co-variability (out of phase) at the 12-month frequency band.

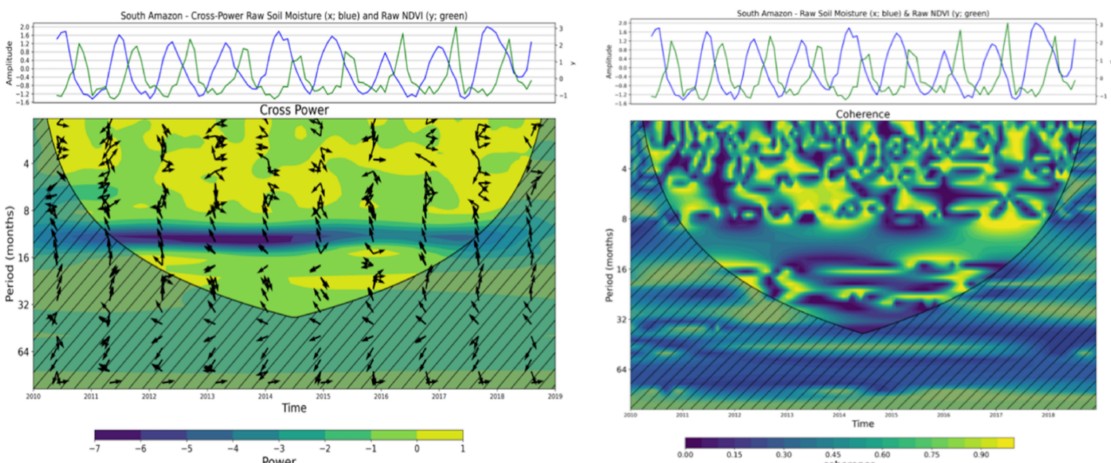

**Figure 6.** Wavelet cross-spectra (WCS, **left** panel) and wavelet coherency analyses (WCO, **right** panels) for the SA region. Arrows pointing to the right in the WCS panels denote that both series are in phase, whereas pointing to the left denote that both series are out of phase. The linear color scale for the WCO results go from deep blue (WCO = 0) to yellow (WCO = 1).

Owing to the definition of the wavelet coherence analyses (WCO), a value of 1 indicates a linear relationship between the two times series at time $t_i$ on a scale frequency $s$, whereas a value of zero indicates null correlation [66]. Results of the WCO analyses shown on the right panel of Figure 6 for all regions indicate that both signals exhibit a strong correlation (yellow) at a wide range of frequency bands, ranging from 2 to 32 months. Cycles shorter than 6 months are abundant and short, possibly owing to the influence of the intra-seasonal or Madden–Julian oscillation on rainfall [67], and the consequent effect on vegetation activity and soil moisture.

Figures S6 and S7 present detailed results of coherence and cross-spectra for the study regions.

### 3.1.5. Nonlinear Causality Analyses

An initial assessment of results allowed us to conclude that the ParrCorr and PCM-CIplus methods produced the most coherent results. It is worth noting that results from the nonlinear causality measures are sometimes difficult to interpret or to link with those obtained from correlation analyses, given that these causality methods can capture nonlinear and even non-monotonic relations among SMOS and NDVI. In general, both nonlinear causality methods confirm the existence of two-way feedbacks between SMOS and NDVI at all regions, as shown in Figure 7, Figures S8 and S9 and discussed next.

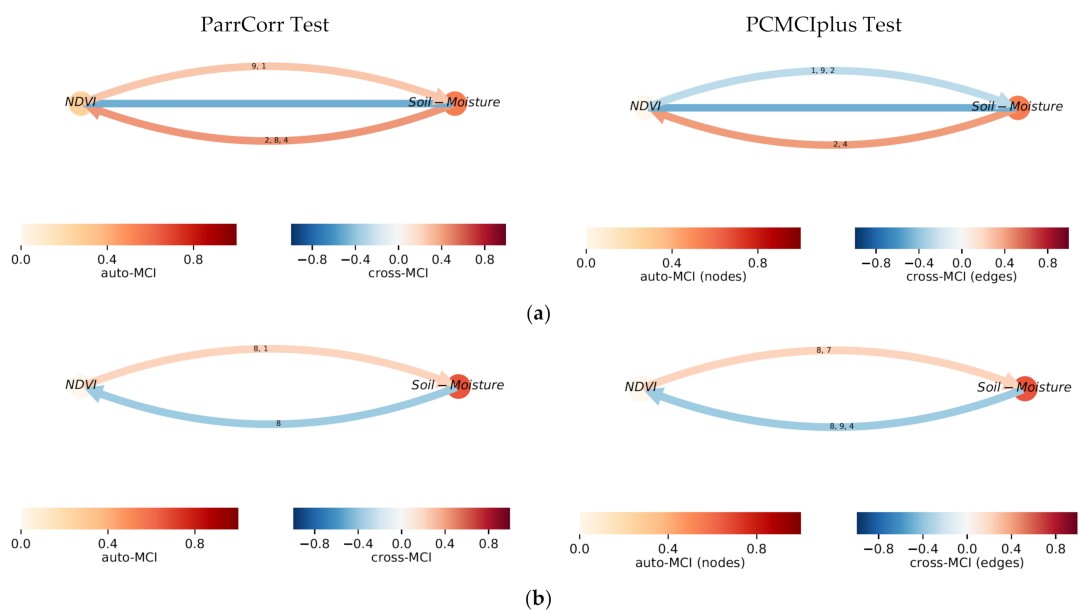

**Figure 7.** Results of the nonlinear causality between NDVI and SMOS for (**a**) Colombia and (**b**) East Amazon: ParrCorr (**left** panels) and PCMCIplus (**right** panels). Node colors denote the nonlinear auto-dependency of each variable, whereas colors of the lines denote the cross-MCI between both variables, and the horizontal lines denote the simultaneous nonlinear co-dependency.

For further interpretation of the figures: node colors denote the nonlinear auto-dependency of each variable, whereas colors of lines denote the cross-MCI between both variables and the horizontal lines denote the simultaneous nonlinear co-dependency.

- ParrCorr test.

NDVI → SMOS. This test identifies significant positive nonlinear causalities in CO (9 and 1-month lags, in that order), and EA (8 and 1-month lags), whereas it identifies negative ones in WA (1-, 4-, 7- and 9-month lags), CA (3- and 5-month lags), and SA (1-month lag).

SMOS → NDVI. This test identifies significant positive nonlinear causalities in CO (2-, 8- and 4-month lags, in that order), WA (5-month lag), and VE (1- and 8-month lags), whereas it identifies negative ones in CA (8- and 5-month lags), EA (8-month lag), and SA (2- and 1-month lags).

Contemporaneous nonlinear co-variability is negative in CO, WA, VE and SA, whereas it is positive in CA and non-significant in EA.

Figure S8 presents the nonlinear causality results for the ParrCorr method. A summary of results is shown in Table S1 of the Supplementary Information.

- PCMCIplus test.

NDVI → SMOS. This test identifies significant positive nonlinear causalities in WA (4-, 7-, and 1-month lags, in that order), EA (8-, and 7-month lags), and SA (7-, 6- and 1month lags), whereas it identifies negative ones in CO (1-, 9-, and 2-month lags), CA (3- and 5-month lags), and EA (8-, 9- and 4-month lags).

SMOS → NDVI. This test identifies significant positive nonlinear causalities in CO (2- and 4-month lags), WA (3-month lag), VE (1- and 8-month lags), and SA (4-month lag), whereas it identifies negative ones in CA (8-month lag), and EA (8-, 9- and 4-month lags).

The contemporaneous nonlinear co-variability is negative in CO, WA, VE and SA, whereas it is positive in CA.

Figure S9 presents the nonlinear causality results for the PCMCIplus method. A summary of results is shown in Table S2 of the Supplementary Information.

Additionally, estimated values of the nonlinear temporal auto-causalities of both NDVI and SMOS allow us to conclude that soil moisture memory is stronger than NDVI, as per

the colors of the nodes in Figures 7, S8 and S9: where the color of auto-MCI varies between 06–0.8 for SMOS and between 0.0–0.6 for NDVI (see Tables S1 and S2).

### 3.2. Standardized Data—Interannual Variability Associated with ENSO
### 3.2.1. Correlation Analyses

- *Lagged cross-correlations between time series*

Lagged cross-correlation analyses are estimated between the standardized times series of monthly regionally averaged values of SMOS and NDVI. The standardization procedure aims to filter out the annual cycle, while keeping the interannual variability of both variables. Toward the end of investigating the effects of both ENSO types, correlations are estimated with the Central Pacific (CP) and Eastern Pacific (EP) indices.

Results indicate that lagged cross-correlations between both ENSO indices and standardized monthly NDVI data are very low, although statistically significant in some cases, and greater for SMOS data. As shown in Table 6, negative (positive) lags indicates that the ENSO index leads (lags) SMOS, therefore, a positive lag means that the variable leads the ENSO index, which we consider physically unlikely, and thus we focus our analysis on the negative lags. Figure S10 presents the correlation results between SMOS and NDVI with the CP and EP indexes, respectively.

**Table 6.** Maximum lagged correlation coefficients and associated time lags between time series of standardized NDVI and SMOS with the CP and EP indices.

| Region | NDVI | | | | SMOS | | | |
|---|---|---|---|---|---|---|---|---|
| | CP | | EP | | CP | | EP | |
| | Max. Corr. | Time Lag (Months) | Max. Corr. | Time Lag (Months) | Max. Corr. | Time Lag (Months) | Max. Corr. | Time Lag (Months) |
| CO | 0.16 | −13 | 0.21 [b] | 0 | −0.72 [a] | −3 | −0.52 [a] | −2 |
| WA | −0.24 [b] | −3 | 0.19 [c] | −13 | 0.38 [a] | 0 | 0.17 [c] | 0 |
| CA | −0.09 | −3 | −0.09 | −7 | −0.35 [a] | −9 | −0.57 [a] | −5 |
| SA | −0.17 [c] | −6 | 0.27 [a] | −3 | 0.22 [b] | 0 | −0.48 [a] | −4 |
| EA | −0.10 | −3 | 0.16 | −3 | −0.47 [a] | −9 | −0.73 [a] | −5 |
| VE | 0.13 | −13 | 0.18 [c] | −1 | −0.62 [a] | 0 | −0.40 [a] | 0 |

Significance levels: [a] 99%; [b] 95%; [c] 90%.

Results for NDVI indicate that the CO, SA, EA and VE regions are more affected by the EP than the CP index, whereas the WA region is most affected by the CP index. Correlations between monthly standardized SMOS data and the ENSO indices exhibit higher and negative values with CP than EP in CO and VE, thus reflecting the negative anomalies of soil moisture in both regions during El Niño [8,23]. The CO, VE, and WA regions have greater correlations with the CP index, so these regions are most affected when SST positive anomalies set in the central tropical Pacific. On the other hand, negative correlations between SMOS and EP are higher for the CA, SA, and EA regions. These results confirm that both ENSO types are associated with different impacts in these regions of tropical South America.

- *Spatial lagged cross-correlations*

Figure 8 shows maps of pixel-by-pixel lagged cross-correlations between monthly values of the CP index and standardized SMOS data, for 0 lag (simultaneous) to 5-month lags. In general, the greatest correlations are seen at CO, VE, WA and SA; being negative at CO, VE and EA, and positive at WA and SA. Correlation values tend to increase for the fourth and fifth lags, and a higher correlation is observed in the SA region. A possible explanation of the inverse sign in the correlation coefficients between CP and SMOS in CO, VE and EA versus those in WA and CA lies in the displacement of the centers of convection within the ITCZ in the eastern Equatorial Pacific towards the south west of their normal

positions [64,68], due to the establishment of an anomalous high pressure center over northern South America, which contributes to explain negative rainfall and soil moisture anomalies over northern and northeastern South America (CO, VE, EA), and positive anomalies to the east of the Andes, mainly over WA and SA [64]. The temporal delay of maximum correlations (4–5 months) could be explained by the westerly travel in time of land–atmosphere anomalies in tropical South America during El Niño, due to the coupling between precipitation, soil moisture and evapotranspiration [8,10,13,14,23,47,51,64,69–73].

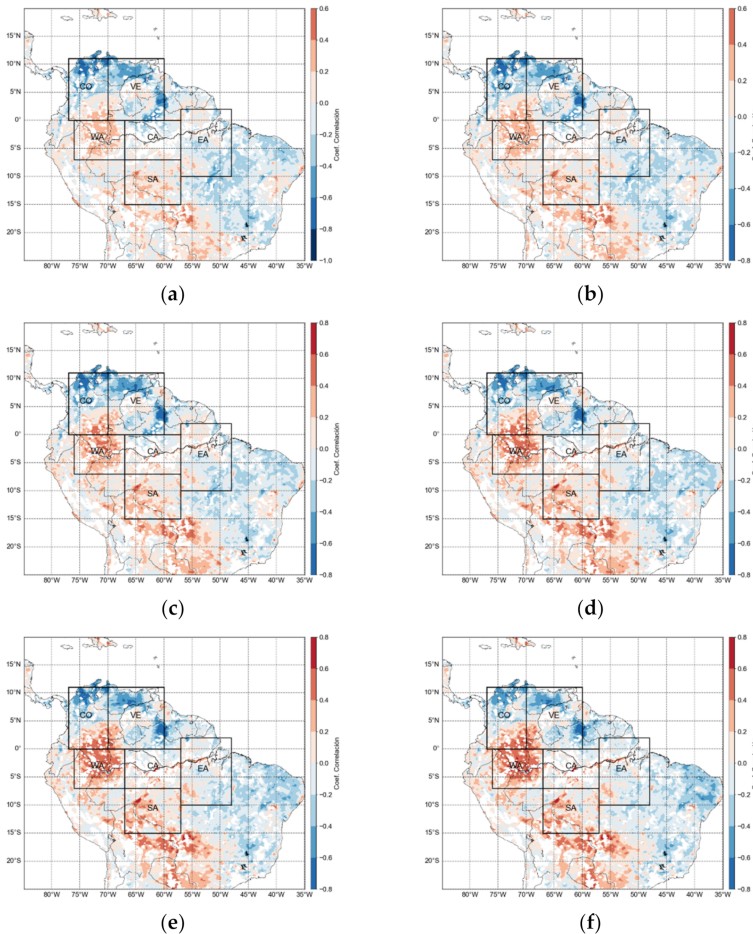

**Figure 8.** Spatial lagged-correlations between the CP index (leading) and standardized SMOS (lagging) monthly data: (**a**) 0 lag; (**b**) 1-month lag; (**c**) 2-month lag; (**d**) 3-month lag; (**e**) 4-month lag; and (**f**) 5-month lag.

Similarly, Figure 9 shows maps of lagged correlations between monthly values of the EP index and standardized SMOS data, for 0- to 5-month lags. In general, the highest correlations appear at CO and VE, being negative in the northern and positive in the southern parts of both countries. However, at 2- and 3-month lags, higher (positive) correlations appear at WA and SA.

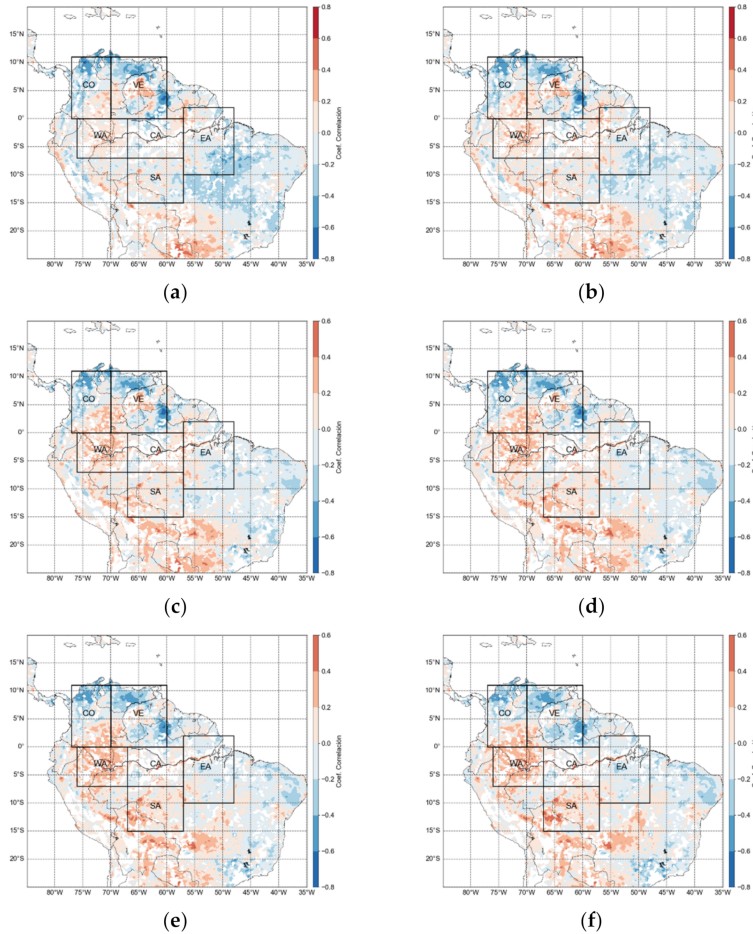

**Figure 9.** Spatial pixel-by-pixel lagged-correlations between the EP index (leading) and standardized SMOS (lagging) monthly data: (**a**) 0 lag; (**b**) 1-month lag; (**c**) 2-month lag; (**d**) 3-month lag; (**e**) 4-month lag; and (**f**) 5-month lag.

It also noted that the CP index presents higher correlations with SMOS (0.2 to 0.8) than the EP index (0 to 0.6).

Maps of lagged correlations allow to identify different types of associations within each region. For instance, Figure 8 shows that lagged correlations between the CP index (leading) and standardized SMOS (lagging) exhibits negative (positive) values over southern (northern) Colombia.

Similar lagged cross-correlation analyses were carried out between monthly values of both ENSO indices and standardized NDVI data, whose results are shown in Figures S11 and S12 of the Supplementary Materials. In general, lagged correlations with NDVI are much smaller than those with SMOS, which can be explained by the complex dynamics of vegetation activity which is mediated by soil moisture and a large suite of biogeochemical processes. Lagged correlations between the CP index and NDVI are almost negligible in all studied regions, whereas with the EP index indicate small, albeit significant positive lagged correlations in CO, VE, CA and SA ($r \approx 0.3$, at 3-month lags).

### 3.2.2. Wavelet Transforms, Wavelet Cross-Spectra, and Coherency Analyses

The most salient result of the wavelet transforms for the standardized data is the strong peak around the 64-month frequency band (WA, CO, VE, SA, CA, EA), in that order and the practical lack of high-frequency variability of soil moisture time series (see Figures 10, S13 and S14). These results confirm that soil moisture exhibits much longer temporal persistence and memory than NDVI. Additionally, NDVI exhibits variability

across a wide range of timescales, which confirms a shorter temporal persistence than soil moisture.

NDVI                                     SMOS

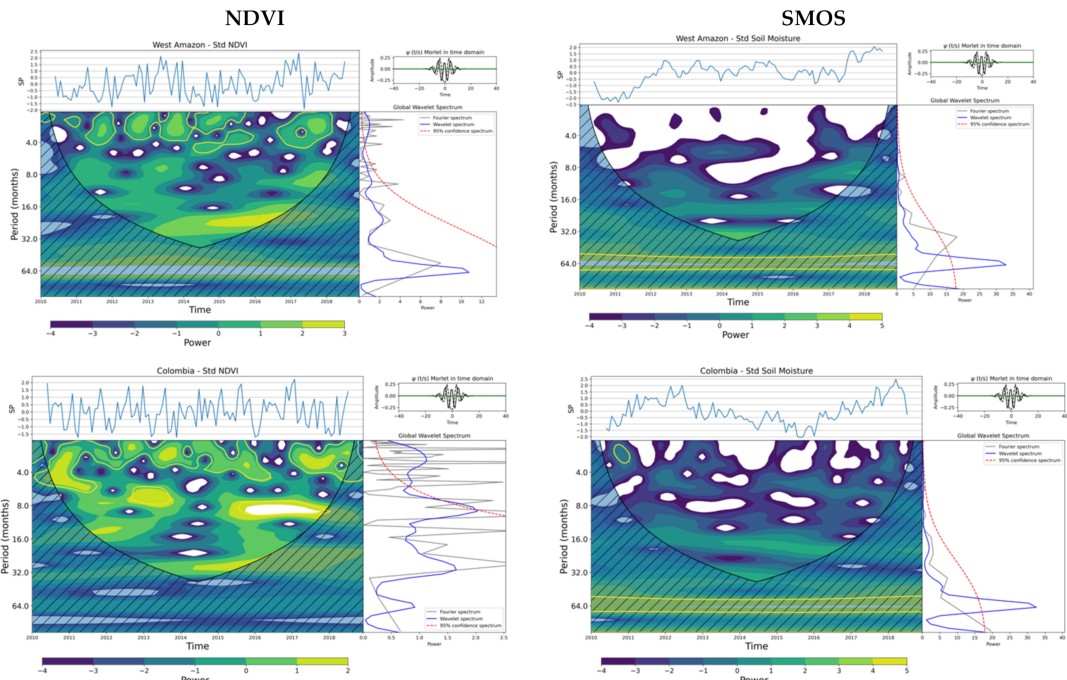

**Figure 10.** Wavelet transforms of monthly standardized series of regional average values of NDVI (**left** panels) and SMOS (**right** panels): (**top**) West Amazon and (**bottom**) Colombia regions.

Results regarding wavelet cross-spectra and wavelet coherency analyses between standardized series of NDVI and SMOS for the six studied regions are shown in Figures S15 and S16 of the Supplementary Materials.

### 3.2.3. Nonlinear Causality Analyses

SMOS, NDVI and Central Pacific (CP)

Figure 11 and Table S3 show results of the ParrCorr nonlinear causality analyses among the monthly standardized series of SMOS and NDVI and the Central Pacific (CP) ENSO index. Figure 12 and Table S4 indicate that similar results were obtained with the PCMCIplus method.

For further interpretation of the figures, it is worth recalling that node colors denote the nonlinear auto-dependency of each variable, while line colors denote the cross-MCI between both variables, whereas horizontal lines denote the simultaneous nonlinear co-dependency.

- *Nonlinear auto-correlations (Auto-MCI).*

In general, estimated values of the nonlinear temporal auto-causalities are higher for SMOS than NDVI, indicating that soil moisture exhibits a stronger temporal memory than NDVI, as per the values (colors) of the nodes in the panels of Figure 11. As expected, significant nonlinear temporal autocorrelations were also found for the CP index, given the strong persistence of ENSO at interannual timescales.

Both nonlinear causality methods present lower results for NDVI auto-MCI at CO, EA, CA and SA regions (0.2 with ParrCorr and 0.1 with PCMCIplus), whereas at WA and VE it reaches to 0.4 with ParrCorr and 0.2 with PCMCIplus.

SMOS auto-MCI results exhibit higher values for all the Amazonian regions up to 0.6, and lower values, around 0.4, for CO and VE with both methods.

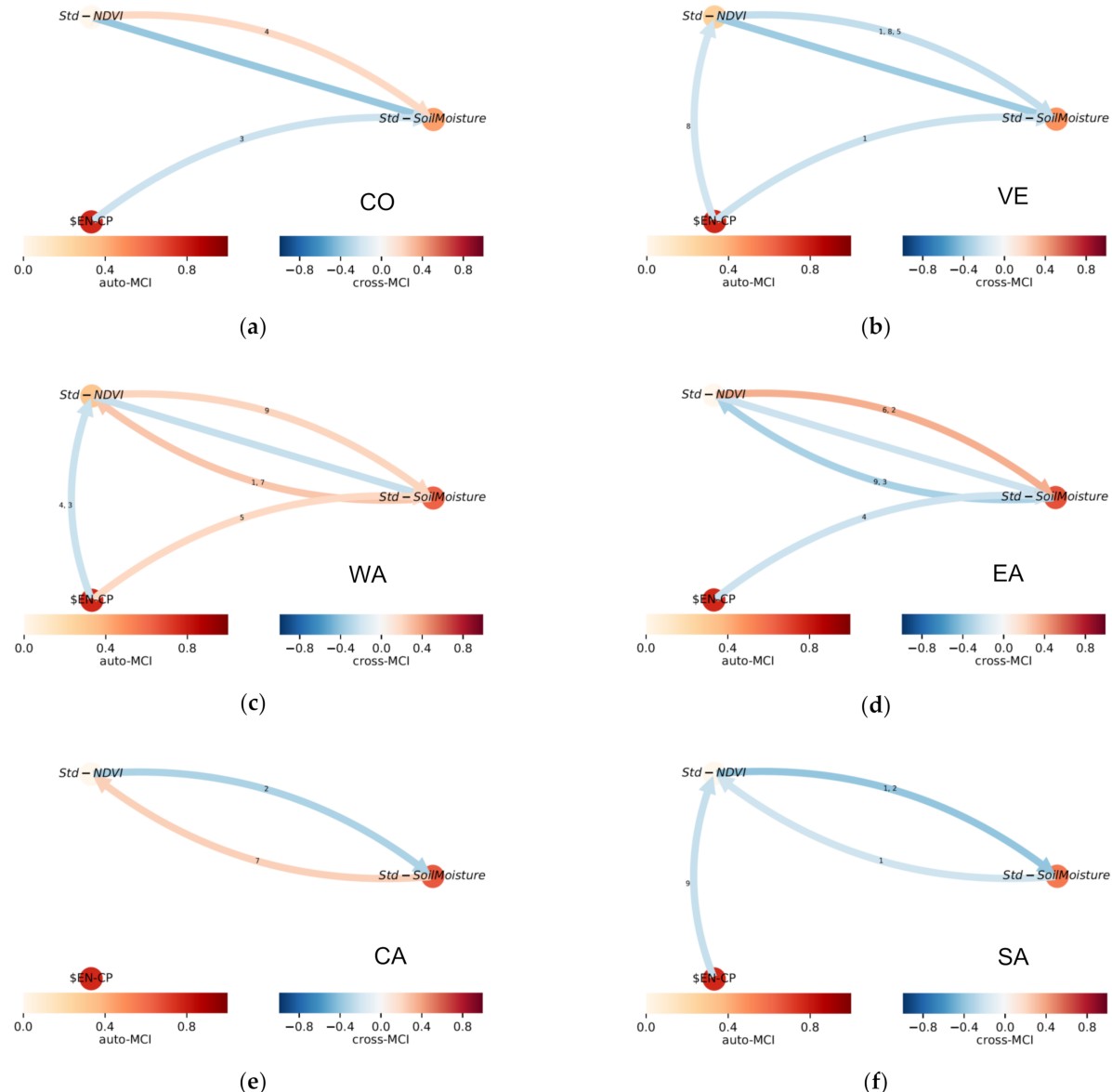

**Figure 11.** Nonlinear causality analysis based on ParrCorr between the standardized monthly series of NDVI and SMOS, and the Central Pacific (CP) index. Node colors denote the nonlinear auto-dependency of each variable, whereas the colors of the lines denote the cross-MCI between both variables and the horizontal lines denotes the simultaneous nonlinear co-dependency. (**a**) CO: Colombia, (**b**) VE: Venezuela, (**c**) WA: West Amazonia, (**d**) EA: East Amazonia, (**e**) CA: Center Amazonia, and (**f**) SA: South Amazonia.

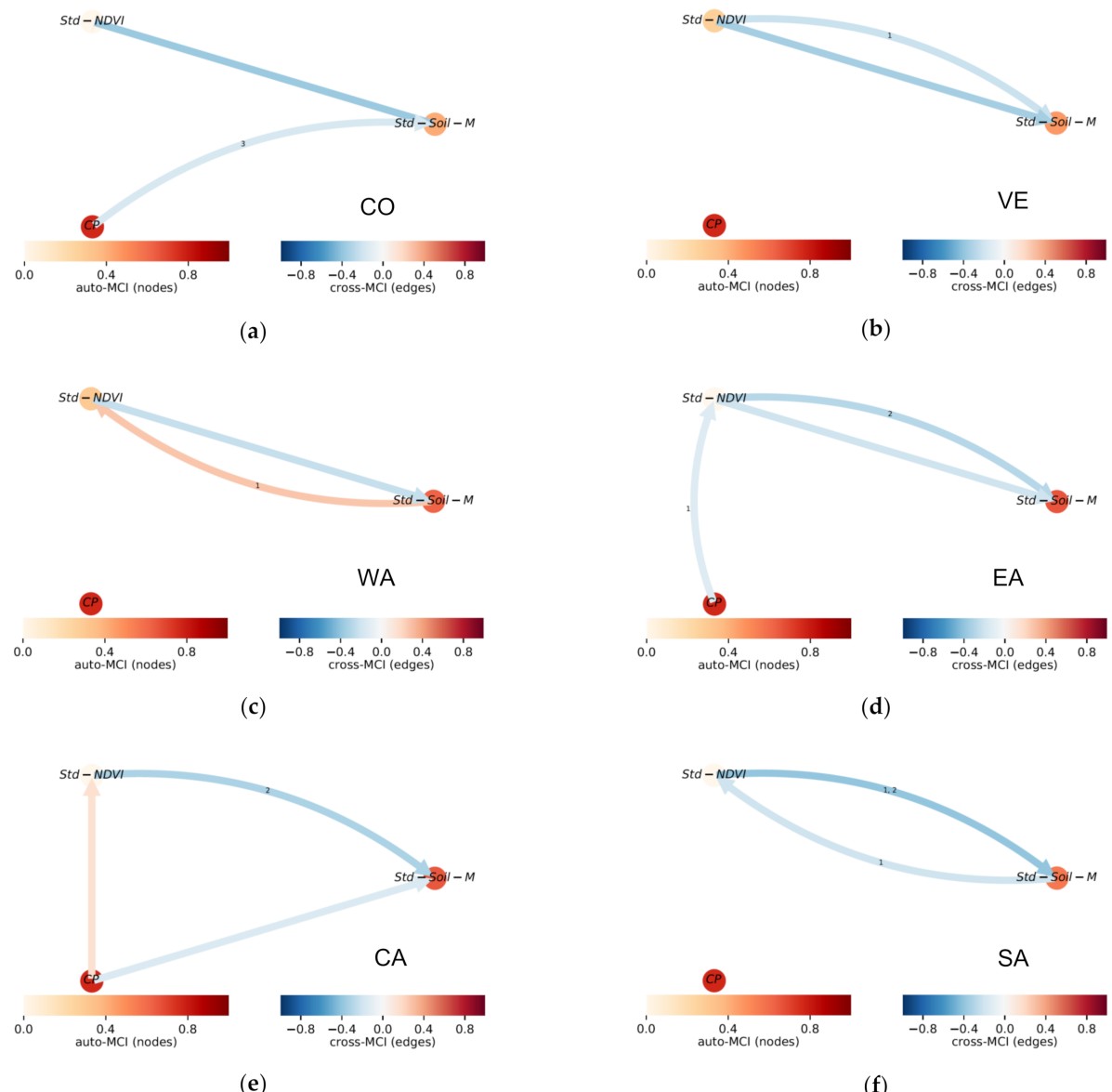

**Figure 12.** Nonlinear causality analysis based on PCMCIplus between the standardized monthly series of NDVI and SMOS, and the Central Pacific (CP) index. Node colors denote the nonlinear auto-dependency of each variable, whereas the colors of the lines denote the cross-MCI between both variables and the horizontal lines denotes the simultaneous nonlinear co-dependency. (**a**) CO: Colombia, (**b**) VE: Venezuela, (**c**) WA: West Amazonia, (**d**) EA: East Amazonia, (**e**) CA: Center Amazonia, and (**f**) SA: South Amazonia.

- *Contemporaneous nonlinear causalities.*

  Both methods indicate that most regions do not exhibit a significant contemporaneous nonlinear causality except for CA (with PCMCIplus), with both variables exhibiting positive nonlinear causalities with the CP ENSO index.

- *Lagged nonlinear causalities.*

  In general, results of the ParrCorr test confirm the existence of causalities between the CP index and both SMOS and NDVI variables for most regions except for CA, as shown in Figure 11 and Table S3. However, the PCMCIplus test only identifies causalities in 2 out the 6 study regions, probably because the PCMCIplus is a more stringent method than ParrCorr, given that it can identify the full, lagged and contemporaneous causal graph (up

to the Markov equivalence class for contemporaneous) under the standard assumptions of Causal Sufficiency, and the Markov condition. It is worth noting that most results indicate negative nonlinear causalities, as follows:

CP → NDVI. The ParrCorr test identifies negative nonlinear causalities in WA (4 and 3-month lags), VE (8-month lag), and SA (9-month lag), and no positive nonlinear causalities were found. The PCMCIplus test identifies negative nonlinear causalities only at EA (1-month lag), and no positive nonlinear causalities were found.

CP → SMOS. ParrCorr test identifies negative nonlinear causalities at VE (1-month lag), CO (3-month lag), and EA (4-month lag), whereas it identifies positive ones at WA (4 and 3-month lag); for CA no nonlinear causalities were found. The PCMCIplus test identifies negative nonlinear causalities only at CO (3-month lag), and no positive nonlinear causalities were found.

Negative causalities identified through both methods imply that negative anomalies of SMOS and NDVI are witnessed during El Niño (positive CP index values), mainly over WA, VE, SA, with a shorter (larger) time delays in those regions closer (farther) to the Pacific Ocean. This result also confirms the aforementioned westerly displacement of ENSO anomalies over the continent, due to precipitation, soil moisture, evapotranspiration feedbacks.

SMOS, NDVI and Eastern Pacific (EP)

Figure 13 and Table S5 show results of the ParrCorr method between the standardized series of SMOS and NDVI and the Eastern Pacific (EP) index. Figure 14 and Table S6 shows similar results for the PCMCIplus method. The main results obtained are as follows:

- *Nonlinear autocorrelations (Auto-MCI).*

Estimated values of the temporal nonlinear autocorrelations of both NDVI and SMOS, confirm that memory of soil moisture is stronger than NDVI at interannual timescales, as per the values (colors) of the nodes in most panels of Figure 14. Additionally, significant nonlinear temporal autocorrelations were found for the EP index, due to the strong persistence and nonlinear memory of ENSO.

- *Contemporaneous nonlinear causalities.*

As for the CP results, both methods indicate that most regions do not exhibit a significant contemporaneous nonlinear causality with the EP index, except for a negative contemporaneous nonlinear causality with EA (ParrCorr) and CA (PCMCIPLUS), both with NDVI. No significant results are found for SMOS.

- *Lagged nonlinear causalities.*

In general, small values are found with both ParrCorr and PCMCIplus tests, with few regions showing significant causalities between the EP index and both SMOS and NDVI, as shown in Figures 13 and 14 and Tables S5 and S6, discussed next.

EP → NDVI. The ParrCorr test identifies negative nonlinear causalities only at WA (8-month lag) and the PCMCIplus test identify small but significant negative nonlinear causalities only at SA (3-month lag). No positive nonlinear causalities are identified with both tests.

EP → SMOS. ParrCorr test identifies negative nonlinear causalities in WA (3-month lag), CA (5-month lag), and SA (4 and 6-month lags). The PCMCIplus test identifies negative nonlinear causalities only at CA (5-month lag), and no positive nonlinear causalities are found. These results also confirm that the effects of ENSO manifest as a traveling wave over time, from northwest (earlier) to southeast (later) over the Amazon River basin. This conclusion confirms the findings of Poveda and Mesa [8] using (linear) correlation analysis.

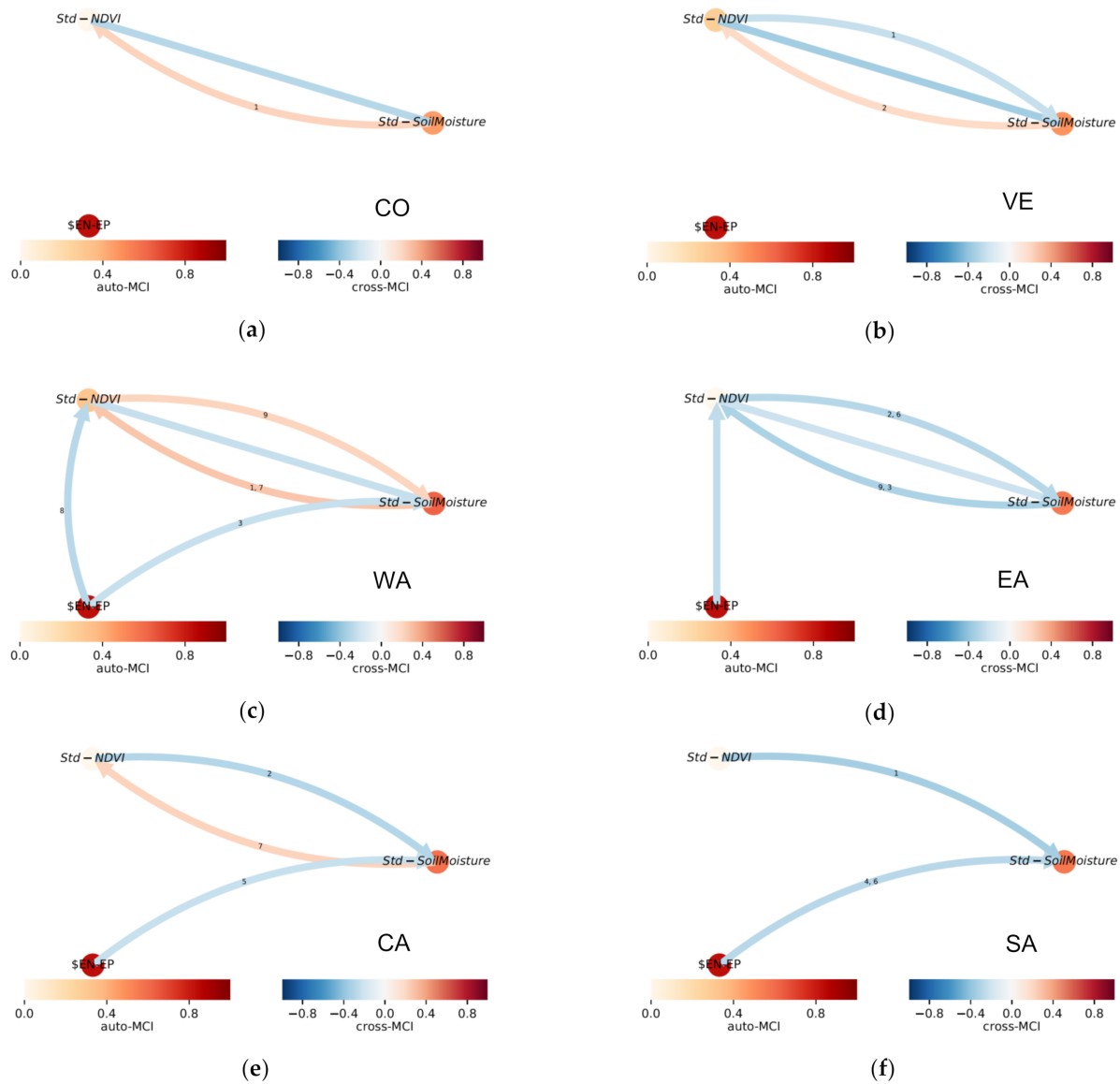

**Figure 13.** Nonlinear causality analysis based on ParrCorr between the standardized monthly series of NDVI and SMOS, and the Eastern Pacific (EP) index. Node colors denote the nonlinear auto-dependency of each variable, whereas the colors of the lines denote the cross-MCI between both variables and the horizontal lines denotes the simultaneous nonlinear co-dependency. (**a**) CO: Colombia, (**b**) VE: Venezuela, (**c**) WA: West Amazonia, (**d**) EA: East Amazonia, (**e**) CA: Center Amazonia, and (**f**) SA: South Amazonia.

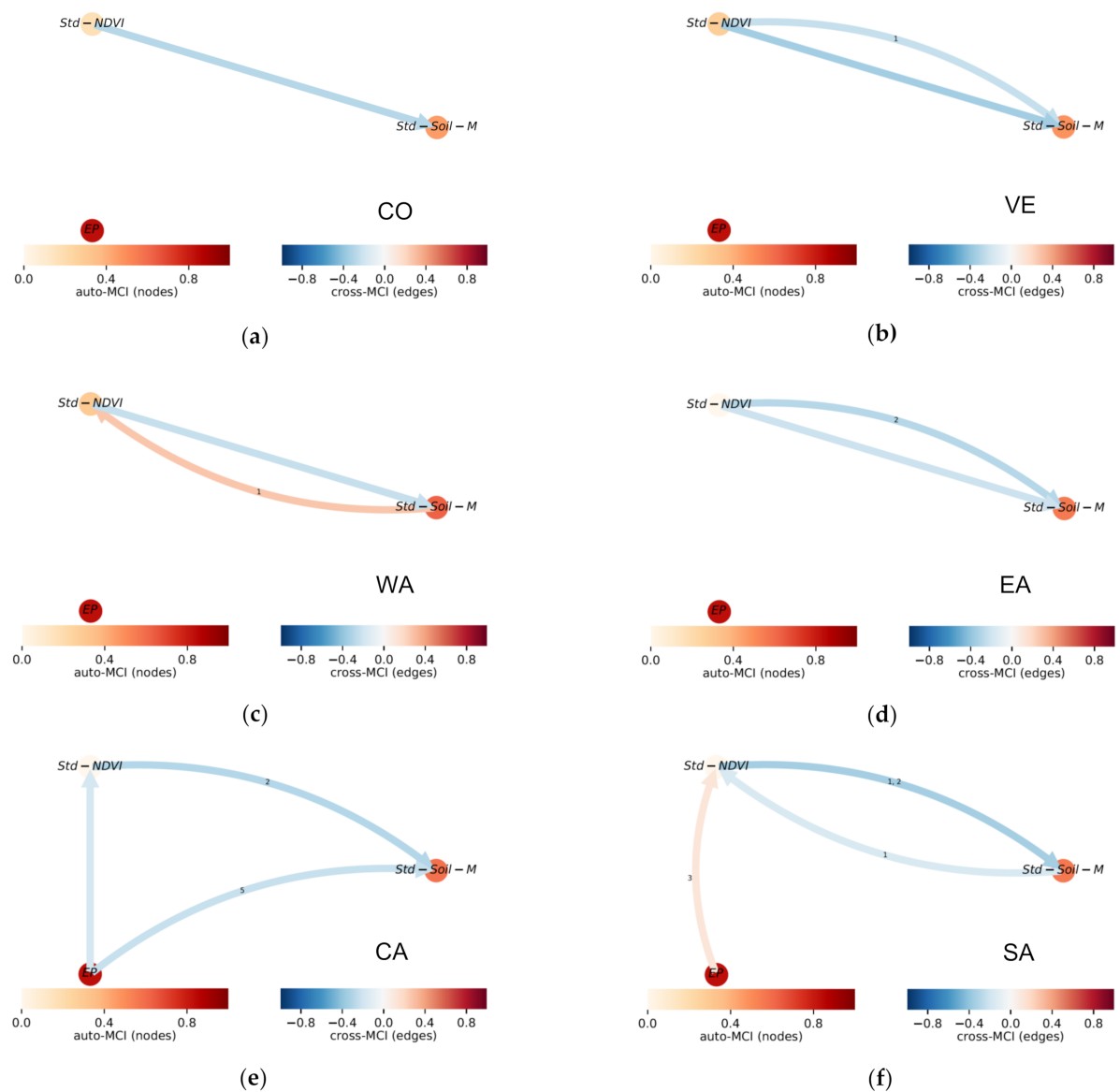

**Figure 14.** Nonlinear causality analysis based on PCMCIplus between the standardized monthly series of NDVI and SMOS, and the Eastern Pacific (EP) index. Node colors denote the nonlinear auto-dependency of each variable, whereas the colors of the lines denote the cross-MCI between both variables and the horizontal lines denotes the simultaneous nonlinear co-dependency. (**a**) CO: Colombia, (**b**) VE: Venezuela, (**c**) WA: West Amazonia, (**d**) EA: East Amazonia, (**e**) CA: Center Amazonia, and (**f**) SA: South Amazonia.

## 4. Discussion

### 4.1. Raw Data

#### 4.1.1. Linear Correlations

Both, annual cycles and the simultaneous time series of monthly data show that the maximum and minimum values of SMOS always precede those of NDVI. This behavior is confirmed by lagged cross-correlation analyses in CO and VE (2 months), in WA, CA and EA (3 months), and in SA (4 months). This southward increasing time delay in the relation between SMOS and NDVI must be further investigated in terms of the relevant biogeophysical processes. On the other hand, NDVI tend to lead SMOS at much longer timescales: 8 months in WA and CA, 9 months in EA and SA, and 10 months in CO and VE. Likewise, this northward increasing time delay in the relation between NDVI and SMOS deserves further investigation.

Additionally, spatial lagged cross-correlations indicate significant positive correlations for most regions, especially when SMOS leads NDVI by 1, 2, and 3 months, in particular, over the Colombian-Venezuelan Llanos of the Orinoco River basin, and smaller negative correlations when NDVI leads SMOS by 2 and 3 months, indicating that the highest vegetation activity (maximum NDVI) occurs when soil moisture is transitioning from maximum to minimum monthly values, which highlights the role of soil moisture on the dynamics of the Amazonian greening up during the dry (precipitation) season.

### 4.1.2. Wavelets and Cross-Wavelet Analyses

Wavelet transforms analyses of all regions show significant peaks at the 12-month frequency band throughout the entire record length, which becomes more evident in the Fourier and the wavelet spectra. The strongest intra-annual variability occurs in Colombia, mostly due to the semi-annual cycle associated with the passage of the ITCZ twice a year, mainly over the Andean region

Time series of SMOS also exhibit an important spectral peak at the 64-month frequency band, mostly associated with ENSO, albeit located inside the nonsignificant cone of influence. Such a 64-month spectral peak only appears in time series of NDVI for the western Amazon.

Wavelet cross-spectra indicate strong positive co-variability at frequencies associated with the annual cycle (12 months) for CO, VE and CA; negative co-variability of both variables around the 12-month frequency band with a small lag (1–2 months) for EA; and a completely different behavior (compared with other regions) at SA, with a strong co-variability (in phase) of both signals at intra-annual timescales, whereas there is a negative co-variability (out of phase) at the 12-month frequency band.

Wavelet coherency analyses evidenced a strong correlation at a wide range of frequency bands, from 2 to 32 months. Cycles shorter than 6 months are abundant and short, possibly owing to the influence of the intra-seasonal or Madden–Julian oscillation on rainfall and the consequent effect on vegetation activity and soil moisture over the study regions.

### 4.1.3. NonLinear Causality Analyses

An initial assessment of results allowed us to conclude that the ParrCorr and PCM-CIplus methods produced the most coherent results and confirmed the existence of both simultaneous and lagged nonlinear causalities between SMOS and NDVI. Most regions exhibit negative simultaneous causalities between both variables after the ParrCorr and PCMCIPLUS methods, with the exception of the CA region (ParrCorr).

Regarding lagged nonlinear causalities, all regions exhibit two-way feedbacks (NDVI to SMOS and SMOS to NDVI), with the only exception of VE (ParrCorr test) where only the causality from SMOS to NDVI is identified.

The first method (ParrCorr) identifies significant positive nonlinear causalities in CO and EA, whereas it identifies negative ones in WA, CA and SA, where NDVI leads SMOS, and significant positive nonlinear causalities in CO, WA and VE, whereas there are negative ones in CA, EA, and SA where SMOS leads NDVI. The second method (PCMCIplus) identifies significant positive nonlinear causalities in WA, EA and SA, whereas there are negative ones in CO, CA and EA where NDVI leads SMOS; and significant positive nonlinear causalities in CO, WA, VE, and SA, whereas there are negative ones in CA and EA where SMOS leads NDVI.

Additionally, estimated values of the nonlinear temporal auto-causalities of both NDVI and SMOS allow us to conclude that the nonlinear memory or persistence of soil moisture is much longer than that of NDVI.

### 4.2. Standardized Data—Interannual Variability Associated with ENSO

### 4.2.1. Correlation Analysis

Temporal correlation analyses indicate high correlations between the CP and EP indices and standardized monthly SMOS data, whereas there were low correlations with

standardized monthly NDVI data, although statistically significant in some cases. Additionally, results indicate that for NDVI, the CO, SA, EA and VE regions are more affected by the EP index than by the CP index, whereas the WA region is most affected by the CP index. Correlations between monthly standardized SMOS data and the ENSO indices exhibit higher (and negative) values with CP than EP in CO and VE, thus reflecting the negative anomalies in soil moisture in both regions during El Niño. The CO, VE, and WA regions show higher correlations with the CP index, so these regions are mostly affected when the anomalous warming of the sea surface temperature occurs in the Central tropical Pacific. On the other hand, negative correlation between SMOS and EP are higher for the CA, SA, and EA regions. These results confirm that both ENSO types are associated with different impacts in these regions of tropical South America.

Spatial correlations analyses indicate that SMOS exhibits the highest correlations at CO, VE, WA and SA; being negative for CO and VE and positive at WA and SA, with correlation values tending to increase for the fourth and fifth lag, and the highest correlation is observed in the SA region. It also noted that the CP index presents higher correlations with SMOS than the EP index.

In general, lagged correlations between the ENSO indices and NDVI are much smaller than those with SMOS, which can be explained by the complex dynamics of vegetation activity which is mediated by soil moisture and a large suite of biogeochemical processes. Lagged correlations between the CP index and NDVI are almost negligible in all studied regions, whereas with the EP index they indicate small, albeit significant positive lagged correlations in CO, VE, CA and SA ($r \approx 0.3$, at lag 3 months).

### 4.2.2. Wavelets and Cross-Wavelet Analyses

The most significant result of the wavelet transforms between standardized monthly series of NDVI and SMOS is the strong peak around the 64-month frequency band, and the practical lack of high-frequency variability of soil moisture at interannual time scales. This result confirms that soil moisture exhibits longer temporal persistence and memory than NDVI. Additionally, NDVI exhibits variability across a wide range of timescales, which confirms a shorter temporal persistence than soil moisture.

### 4.2.3. Nonlinear Causality Analyses

Nonlinear auto-correlations results indicate that for both indices and both tests, SMOS exhibited higher values than NDVI, indicating that the nonlinear temporal persistence (memory) of soil moisture is stronger than NDVI. Significant nonlinear temporal auto-causalities were found for both indices.

Contemporaneous nonlinear causality results (ParrCorr and PCMCIplus tests) indicate that the ParrCorr test confirms the existence of higher lagged nonlinear causalities between the CP index and both variables for most regions but CA, whereas there are smaller values or non-existent causalities for the EP index.

### 4.3. Study Limitations

One of the limitations of the present study is the short record length, constrained by the available satellite soil moisture data set (January 2010 to December 2018), in particular to understand the dynamics at interannual (ENSO) timescales. Other studies must involve longer data sets to disentangle the linkages between the hydrologic cycle and vegetation activity in the study regions [74–77].

Another source of uncertainty is the quality of satellite soil moisture data set. Rigorous ground-validation studies will be required to assess their representativeness at the spatiotemporal scales of interest. For instance, the Pacific coast of Colombia has been identified as the rainiest region on Earth, with long-term mean annual rates reaching 13,000 mm [78–81]. The region is mostly covered by tropical rainforest, and yet SMOS exhibits lowest values, similar to those found over the dry north-east region of Brazil, which seems highly unlikely. One possible explanation is the permanent presence of large

amounts of cloudiness associated with the year-long occurrence of Mesoscale Convective Systems over the region, which could be introducing noise and distortion to the satellite measurements and causing the high number of missing SMOS records over the region.

## 5. Conclusions

We have studied the conjoint dynamics of soil moisture (SMOS), NDVI and ENSO in six regions of tropical South America, using diverse linear and nonlinear methods. Analysis of the raw data indicated that the maximum and minimum values of SMOS led those of NDVI by 2–4 months for all regions, with the lag exhibiting an increasing southward trend in the relation between SMOS and NDVI. At the same time, results also showed that NDVI tends to lead SMOS at much longer timescales (8 to 10 months), with the lag exhibiting a northward trend. These results deserve further investigation in terms of the relevant biogeophysical processes. In terms of the long-term mean annual cycle, results showed that maximum monthly values of NDVI occur when soil moisture is transitioning from maximum to minimum monthly values, thus pointing out the role of soil moisture on the identified greening up of Amazonia during the dry (precipitation) season.

Wavelet analyses of the raw data showed significant peaks at the 12-month frequency band throughout the entire record length, which became more evident in the Fourier and the wavelet spectra. The strongest intra-annual variability occurred in Colombia, mostly due to the semi-annual cycle associated with the passage of the ITCZ twice a year, mainly over the Andean region. Time series of SMOS also exhibited an important spectral peak at the 64-month frequency band, mostly associated with ENSO, albeit located inside the non-significant cone of influence. Such a 64-month spectral peak only appeared in time series of NDVI for the western Amazon. The most significant results of the cross-wavelet analysis between standardized monthly series of NDVI and SMOS was the strong peak around the 64-month frequency band, and the practical lack of high-frequency variability of soil moisture at interannual time scales. This result confirmed that soil moisture exhibited longer temporal persistence and stronger memory than NDVI. Additionally, NDVI exhibited variability across a wide range of timescales, which confirmed a shorter temporal persistence than soil moisture.

As with the linear analyses, the nonlinear causality methods demonstrated the existence of both simultaneous and lagged nonlinear two-way causalities between SMOS and NDVI, with the only exception of VE (ParrCorr test) for which the causality from SMOS to NDVI was identified.

Linear correlation analyses also indicated high values between the CP and EP indices of ENSO and standardized monthly SMOS data, whereas low correlations with standardized monthly NDVI data, although statistically significant in some cases. Differences identified among the diverse study regions confirmed the pertinence of such regionalization across northern South America, given their differences in location, latitudinal range, vegetation cover, and hydrological dynamics at annual and interannual timescales, such that some regions appeared more associated with the EP index, whereas there were others with the CP index. Correlation analyses of time series of regionally averaged values of SMOS and NDVI, as well as spatial (pixel-by-pixel) correlation analyses showed that the CP index exhibited higher correlations with SMOS than the EP index and, in general, lagged correlations between the ENSO indices and SMOS were much higher than those with NDVI, which might be explained by the complex dynamics of vegetation activity, which is mediated by soil moisture and a large suite of biogeochemical processes.

The ParrCorr and PCMCIplus methods produced the most coherent results among the nonlinear causality methods, and allowed us to conclude that: (1) the nonlinear temporal persistence (memory) of soil moisture is stronger than that of NDVI; (2) the existence of two-way nonlinear causalities between NDVI and SMOS, and (3) diverse causal links between both variables and the ENSO indices: CP (7/12 with ParrCorr; 6/12 with PCMCIplus), and less with EP (5/12 with ParrCorr; 3/12 with PCMCIplus).

Further analyses are in order to corroborate and complement these results using longer record lengths, and much better validated data sets, in particular of SMOS, for which rigorous ground-validation studies will be required to assess its representativeness at the spatiotemporal scales of interest.

**Supplementary Materials:** The following supporting information can be downloaded at: https://www.mdpi.com/article/10.3390/rs14112521/s1, Figure S1: Location of the distinctive regions along the tropical Pacific commonly used to study the dynamics of ENSO; Figure S2: Chart flow of the methodology used in the study; Figure S3: Time series of monthly values of NDVI and soil moisture for the study regions; Figures S4–S7 and S13–S16: Further results of the wavelets analyses, wavelet cross-spectra analyses, and coherency analyses for the study regions; Figures S8 and S9: Detailed results of nonlinear causality analyses; Figures S10–S12: Further results of correlation analyses between soil moisture, NDVI and ENSO indices; Tables S1 and S2: Nonlinear causality correlations and causality results for the ParrCorr and PCMCIplus methods; Tables S3–S6: Nonlinear correlations and causality results between the ENSO indices (CP and EP) with the different causality methods.

**Author Contributions:** Conceptualization: G.P.; Data curation: D.M.Á.; Formal analysis: D.M.Á. and G.P.; Investigation: D.M.Á. and G.P.; Software: D.M.Á. and G.P.; Validation: G.P.; Visualization: D.M.Á. and G.P.; Writing: D.M.Á. and G.P. All authors have read and agreed to the published version of the manuscript.

**Funding:** This research received no external funding.

**Data Availability Statement:** The datasets used for the analysis in Section 2.2.1. is freely available at http://bec.icm.csic.es/land-datasets/, accessed on 25 August 2019. The datasets used for the analysis in Section 2.2.2. is freely available at https://doi.org/10.7289/V5ZG6QH9, accessed on 25 August 2019. The code source used in Section 2.3.2. is freely available at https://github.com/mabelcalim/waipy, accessed on 20 January 2020. The code source used in Section 2.3.3. is freely available at https://github.com/jakobrunge/tigramite/, accessed on 20 March 2020.

**Acknowledgments:** G.P. acknowledges Universidad Nacional de Colombia at Medellín, Colombia for support. We thank Mabel C. Costa and Jakob Runge for making available the *waipy* and *Tigramite* packages, respectively.

**Conflicts of Interest:** The authors declare no conflict of interest.

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
