# Peer review of "Spatiotemporal Dynamics of NDVI, Soil Moisture and ENSO in Tropical South America"

_remotesensing, doi:10.3390/rs14112521_

Round 1

Reviewer 1 Report

Inappropriate citation. please maintain one reference system

No chapter on DISCUSSION. This is a great ommission.

Mix up of tenses

See my attached comments

Author Response

Dear reviewer, Please see the attachment to a point-by-point response, we have made all efforts to improve the manuscript by attending all your comments.

Thank you in advance.

Reviewer 2 Report

Dear author,

I think you have made a substantial revision to this draft. Even though the manuscript still feels a little bit too long but it is far more readable than the previous draft. some minor comments:

  1. add workflow to the manuscript, so that the processing flow can be better described
  2.  add a  brief conclusion to summarize the study's findings, the present could be transformed into a discussion part

Author Response

Dear reviewer, please see the attachment to a point-by-point response, we have made all efforts to improve the manuscript by attending all your comments.

Thank you in advance.

Reviewer 3 Report

Thank the authors for their comprehensive answers;

The content on page 6 (line 215, 228) and 7 (line 237) (manuscript_version1) is not clear in places - have been changed in the current version of the manuscript

Author Response

(The authors gave the same response as above.)

Round 2

Reviewer 1 Report

All my comments have een addressed. You can now publish after minor spelling mistakes.

Author Response

Thank you so much for all your observations, we improve the English language as you recommended and included some new references [89-92] in the Discussion section.

We are positive that the quality of the manuscript has been greatly improved thanks to the Editor's and the reviewers’ comments. 

This manuscript is a resubmission of an earlier submission. The following is a list of the peer review reports and author responses from that submission.

Round 1

Reviewer 1 Report

This is the second time I reviewed this manuscript. I personally believe the study is important and interesting. However, my main concern still lies in the crowded information within a manuscript which makes the results are difficult to be followed. In addition, the Introduction is now added but lack of systematic structure and no research gap are presented. It is unfortunate that I still have to recommend rejection to this manuscript until the writing is revised.

Other comments are as followed:

Introduction

  1. Line 49 – 84, please separate this paragraph into two paragraphs since this is too long for the reader to follow
  2. What is the research gap? Please state in the introduction. Right now I can not understand your justification for conducting this study.
  3. Please state the reason for choosing NDVI as the proxy of vegetation dynamics especially if you use the time period of 2010 – 2018 where plenty of vegetation proxies from remote sensing data are available

Materials and methods

  1. “For our purposes, this is a continental-scale study area because of its important role in regional (and global) climate and hydrology, particularly the Amazon River basin” this sentence is confusing please rephrase
  2. Why did you divide the Western, central, eastern, and southern Amazonia? Do you have specific boundaries of those regions? Why don’t you use the whole region for the analysis?
  3. Please make the study area map to better explain your region.
  4. For the wavelet data, what kind of mother wavelet that you used? And please justify
  5. The time-series data are bound to be non-stationary data. Please justify why don’t you transform the data into a stationary time series before conducting the analysis?

Results and discussion

  1. “Results indicate that the maximum and minimum values of SMOS tend to lead those of NDVI by 2-4 months, depending on each region” Do you conduct any quantitative analysis to justify this statement?
  2. Still, too many results make this study hard to be read. It is better to separate this into two manuscripts or to reduce the regions studied in this manuscript (or merge them into one region)

Reviewer 2 Report

My concern was mostly addressed, and this study is worth publishing.

There is a minor comment for Figure 5: Text in the vertical label is broken.

Reviewer 3 Report

The reviewed manuscript is interesting, but there is requires:

- there are many abbreviations in the text of the manuscripts, including the names of regions, which discourages for reading - I suggest using fewer abbreviations (although they are explained), the manuscript should be enriched with physical interpretations of the results obtained,

- e.g. in Figure 5,6 the text should be enlarged, currently it is difficult to read

- a lot of results are given in the manuscript, but the presentation in a clear, understandable way is missing, and their interpretation is also advisable - for example, "Figure 13 and Table S5 show the results of the ParrCorr method between the standardized series of SMOS and NDVI and the Eastern Pacific (EP) ENSO index. Figure 14 and Table S6 shows similar results for the PCMCIplus method” (page 17 line 494-496)

- “One of the limitations of the present study is the short record length, constrained by the available satellite soil moisture data set (2010–2019)” - I suggest checking the given period with the information provided in Table 2 and in point 2.2.1 (page 3 line 105),

- explanation of the reasons for the delay, eg: “EP → SMOS. ParrCorr test identifies negative non-linear causalities in WA (3-month lag), CA (5-month lag), and SA (4 and 6-month lag). The PCMCIplus test identifies negative nonlinear causalities only at CA (5-month lag), and no positive non-linear causalities are found: “

Comments for Authors

Page 2 line 47-48

"... the extreme drought of 20005 deep root uptake occurred at greater depths as a mechanism to cope with prolonged droughts." – the notation is not clear,

Page 3 line 96

I suggest providing a brief description of the research area,

Page 3 line 97-98

"For our purposes, this is a continental-scale study area because of its important role in regional (and global) climate and hydrology, particularly the Amazon River basin [39-43]." - the notation is not clear, it requires an explanation

Page 3 line 103

It should be clearly indicated which data were included in the study, for which years?

Page 3 line 126-128

"From the monthly data it is possible to identify possible correlations and effects between the variables under study, allowing the identification of variations and timescales associated with macroclimatic events such as the ENSO phenomenon." - the notation is not clear, requires complementing. Pay attention to the variables - which ones?

Page 4 line 135

"The Niño 3 and Niño 4 regions…" - requires explanation;

Figure S1. Sea surface temperature in the Equatorial Pacific in presence of the El Niño (La Niña) phenomenon ”- Figure S1 does not show temperature - what are the temperature values, in which unit the temperature is given – Figure is not clear.

Page 4 line 154

“2.3.1. Exploratory analysis ”- chapter is too short - I suggest adding the information to another one

Page 4 line 159

“For both raw and standardized times series and temporal and spatial data… ”- the notation is not clear. If spatial data are not related to time? – requires explanation.

Page 6 line 228

"Results indicate that the maximum and minimum values of SMOS tend to lead those of NDVI by 2-4 months, depending on each region" - I suggest rising to the notation: "For both variables, location indices are similar; therefore, it can be said that both series lack extreme values "(Page 6 line 215) and "These plots also confirm that the maximum and minimum values of SMOS always precede those of NDVI "(Page 7 line 237) - the information is not clear

Page 7 Figure 2

I suggest the figure inside removing - figure is not legible

Sign the remaining figures with the name of the region and add one legend for the entire Figure2

Page 7 line 249

“Significance levels: (a) 99%; (b) 95%; (c) 90% "- only (a) is given in the table - I suggest complementing the table with the significance level (b), (c) or removing

Page 8 Figure 3

I suggest the figure inside removing. This drawing is not legible.

Page 8 Figure 4

The drawings are not legible - I suggest increasing them and providing the description of the results and their appropriate quotation in the text

Page 10 line 320

"... and non-monotonic relations among both variables" - I suggest listing these variables

Page 12 line 369

"Lagged cross-correlation analyzes are estimated using the standardized times series ..." - for the sake of clarity of the text, I suggest specifying what the time series refer to

Page 13 line 397-400

“Figure 8 shows the lagged cross-correlations between monthly values of the CP index and standardized SMOS data, for 0 (simultaneous) to 5-months lags. In general, the greatest correlations are seen at CO, VE, WA and SA; being negative for CO and VE and positive at WA and SA. Correlation values tend to increase for the fourth and fifth lag, and a higher correlation is observed in the SA region "- I suggest adding what could be the cause and what is the result of it

Page 16 line 468

"However, the PCMCIplus test identifies causalities in 2 out the 6 study regions" – it is requires adding what are the causes ...?

Page 13 Figure 8

Page 14 Figure 9

The drawings are not legible - the scale is not visible

Page 16 line 469

"It is worth noting that most results show negative non-linear causalities ..." – why? What could be the reason? Comment is requires.

Page 16 Figure 11

Page 17 Figure 12

Page 18 Figure 13

Page 19 Figure 14

The drawing inside is not legible, I suggest removing it.

Page 21 line 626

“One of the limitations of the present study is the short record length, constrained by the available satellite soil moisture data set (2010-2019)” - I suggest checking the given period with the information provided in Table 2 and in section 2.2.1 (page 3 line 105),

Reviewer 4 Report

Dear Authors, 

I have reviwed your manuscript  "Spatiotemporal Dynamics of NDVI, Soil Moisture and ENSO in Tropical South America" and, in my opinion it is interesting  and  sound. But I have few comments.

General comment : it is too long. There are too many analysis what make complete text unclear. But I do not insist on its reduction

Specific comments:

  1. What is  novelty of the research? Please write it clearely.
  2.  Please write sound and clear conclusion. Consclusion written in several  chapters and sub-chapters is not easy to follow.
  3. Please refer to spatial scale of research. It covers large /continental area. How does it reflect on reliability of the research and  acchieved results ?
  4. Line 47 . What does it mean :20005?

Reviewer 5 Report

MANUSCRIPT NUMBER - 1553248-PEER-REVIEW-V1

Title: Spatiotemporal Dynamics of NDVI, Soil Moisture and ENSO in Tropical South America

Authors:.Diana M. Álvarez and Germán Poveda

  1. The abstract:

-This is well done up to th econclusion. Please add a sentence to conclude the study.

  1. The introduction:

This section is too short for an article. It can be enhanced.

Line 47, correct typo error  ‘’20005’’ to read 2005

Delete lines 90-93 which shows the flow of chapters. Instead you could provide a hypothesis of the study.

  1. Materials and methods:
  • In line 102, you could briefly explain which variables occur in which region(s).
  • Lines 149 - 152 can be written in a continous flow. Remove bullets from an article.
  • Methods are clearly explained

  1. Results:

Well presented in detail.

  1. Discussion

No discussion presented. This is not right. Please present discussions for the study.

  1. Conclusion

Well done.

  1. Recommendations

None presented. It is good to invite future studies to improve your results.